# An integrated three-tier trust management framework in mobile edge computing using fuzzy logic

Merrihan B.M. Mansour[1], Tamer Abdelkader[2], Mohamed Hashem[2] and El-Sayed M. El-Horbaty[2]

[1] BAEPS, The British University in Egypt, Egypt
[2] Faculty of Computers & Information Sciences, Ain Shams University, Cairo, Egypt



## ABSTRACT

Mobile edge computing (MEC) is introduced as part of edge computing paradigm, that exploit cloud computing resources, at a nearer premises to service users. Cloud service users often search for cloud service providers to meet their computational demands. Due to the lack of previous experience between cloud service providers and users, users hold several doubts related to their data security and privacy, job completion and processing performance efficiency of service providers. This paper presents an integrated three-tier trust management framework that evaluates cloud service providers in three main domains: Tier I, which evaluates service provider compliance to the agreed upon service level agreement; Tier II, which computes the processing performance of a service provider based on its number of successful processes; and Tier III, which measures the violations committed by a service provider, per computational interval, during its processing in the MEC network. The three-tier evaluation is performed during Phase I computation. In Phase II, a service provider total trust value and status are gained through the integration of the three tiers using the developed overall trust fuzzy inference system (FIS). The simulation results of Phase I show the service provider trust value in terms of service level agreement compliance, processing performance and measurement of violations independently. This disseminates service provider's points of failure, which enables a service provider to enhance its future performance for the evaluated domains. The Phase II results show the overall trust value and status per service provider after integrating the three tiers using overall trust FIS. The proposed model is distinguished among other models by evaluating different parameters for a service provider.

## INTRODUCTION

Cloud computing (CC) provides a variety of computing resources such as processing capabilities, storage, servers, for multiple cloud users over the network (*Monir et al., 2015*). Such resources are physically located at large data centers which are far away from users' proximity. This causes high data transfer delays between service users and cloud resources, resulting in an increased network latency, while preventing real time applications like vehicular networks from being processed in a timely manner (*Roman, Lopez & Mambo,*

Corresponding author
Merrihan B.M. Mansour,
merrihan.mansour@bue.edu.eg

*2016*; *Mach & Becvar, 2017*; *Shi et al., 2016*; *Taleb et al., 2017*). Mobile edge computing had emerged as part of the cloud computing paradigm, in an attempt to be nearer to user premises, under the coverage of radio access networks (RAN) (*Ahmed & Ahmed, 2016*; *Aslanpour et al., 2021*).

In mobile edge computing (MEC), service executions such as computation and storage are transferred from the cloud network to the mobile base stations located at the network edge (*Wang et al., 2017*; *Hu et al., 2015*; *Leppanen, 2019*; *Nunna et al., 2015*). This provided low network latency, scalability and utilization of resources, which in return, minimizes computational and network overhead during data offloading for computational purposes. On the other hand, it enabled real time and data sensitive applications, such as smart health care systems, to be efficiently executed within their time limitation (*Shangguang et al., 2019*; *Chen et al., 2016*; *Shi & Dustdar, 2016*; *Corcoran & Datta, 2016*). MEC allowed more service providers and users to connect to the network, benefiting from the processing and storage capabilities, which became more accessible to them (*Shi, Sun & Cao, 2017*; *Maoy, You & Zhang, 2017*; *Rani et al., 2021*). This had relatively increased the transactions rate and number of participants connected to the MEC paradigm.

## Problem statement

However, due to the large number of communicating entities, several security and trust issues arise in such a vulnerable environment (*Tang & Alazab, 2017*). These security threats such as; fake service users, malicious service providers or denial of service attack (*Jhaveri et al., 2018*). Trust issues arise when service users route their private data for computational purposes, to unknown remote service providers, where they lose control of it (*Sheikh et al., 2012*). Due to lack of previous experience between service users and providers, service users hold several doubts like:

- their data security, confidentiality and privacy (*Deepa et al., 2020*; *Ranaweera, Jurcut & Liyanage, 2021*);
- unknown service provider's processing performance efficiency and trust degree;
- no guarantee that the selected service provider would abide to the agreed upon service level agreement (SLA) terms;
- no recording of historical violations committed by a service provider, and the type of it. This may give a chance for malicious service providers to re-do their incorrect actions again, knowing that they are untraced by any authorized entity.

A service level agreement acts as a contract, signed between a service provider and user that states the agreed upon processing conditions. However, there is not a standard format for an SLA, which obscures its legal judgment. On the other hand, trust data extraction is a very difficult task due to the large number of transactions follow, in which a huge amount of data is generated like transaction type, terms, cost and service users' ratings. Service user ratings, could be untrusted, biased, irrelevant or difficult to filter. Therefore, service users demand guidance of service providers' trust degree prior their

selection. Several works had been introduced in literature that evaluated service providers in terms of processing performance, processing quality, response time or SLA compliance degree. However, to the best of our knowledge at this time, none of the previous works covered all these major parameters together. Some of these works faced challenges such as depending on service users' feedback opinion, lack of trust results update, unclear service provider assessment criteria.

## Motivation

It is essential to build a trust evaluation scheme to evaluate service providers' performance in the MEC environment for four main reasons:

1. Service users will be aware service provider's trust degree prior their interaction.
2. Service providers will understand that their actions are being monitored and recorded in a historical database. This motivates them to enhance their processing performance capabilities and limits any malicious actions to happen.
3. The trust evaluation scheme allows a service provider to know its faulty points to improve them, (*Asghar et al., 2020*).
4. Service providers with good trust value will attract more service users, which increases their profits.

A standard and universal trust evaluation model for MEC entities, would greatly contribute in distinguishing trustworthy service providers among others in the MEC network and their offered services. This would avoid attacks such as, malicious or fake service providers, and collusion attacks. Building trusted relationships would secure future interactions in the MEC paradigm. Consequently, service users' confidence and reliability on the MEC services will increase, leading to higher transactions rate (*Chong et al., 2013a*).

## Paper contribution

Trust is defined as the level of service user confidence towards a service provider for fulfilling its computational requirements as expected (*Chahal & Singh, 2015*; *Ruan, Durresi & Alfantoukh, 2016*; *Ruan & Durresi, 2016*; *Ruan & Durresi, 2017*). A trust management system builds trusted relationships between the participating entities, by assessing each service provider provisioned services and making trust level results available to service users when requested. Therefore, service providers' trust history should be captured, to avoid trust computation prior each new interaction, which saves time and yields to users' awareness of service provider's past interactions.

To address the above limitations, this research introduces the need of a unified trust management framework that evaluates service providers' provisioned services in the MEC network considering various parameters. Trust evaluation is performed in a centralized manner, by a fully trusted third party known as cloud service manager (CSM), to grantee trust results credibility (*Felix & Ricardo, 2012*; *Hatzivasilis et al., 2020*). This also promotes for a secure and successful transactions between service users and providers in the MEC environment (*Rathee et al., 2017*).

On the other hand, fuzzy logic concept is used to address a situation of partial truth or uncertainty of values (*AbdelKader, Naik & Nayak, 2011*). This is an ideal choice, when evaluating the trustworthiness of a service provider, were the trust result is a dynamic variable depending on several measured parameters computed per computational interval (*Tariq et al., 2020*). This paper presents an integrated three-tier trust management framework using fuzzy logic. The main contributions of this paper are:

**Phase I: Three tiers**

1. Evaluation of service provider SLA compliance degree in Tier I;
2. Computation of service provider processing performance in Tier II;
3. Measurement of service provider violations in Tier III;
4. Each tier trust evaluation is performed Three tiers independently per transaction in a batch processing manner.

**Phase II: Three-tiers integration**

5. A MATLAB based overall trust fuzzy inference system (FIS) was developed to integrate the evaluated results of tiers I, II and II, in order to gain an overall trust value and status for a service provider.

In the proposed framework, an SLA trust value is evaluated using four parameters (execution time, storage, cost and maintenance), to have a standard format, which allows it for a consistent judgment (*Sheikh, Sebestian & Max, 2011*). On the other hand, the processing performance of a service provider is measured by computing the number of successful processes verses the total number of accepted jobs, while gaining the failure ratio. The violations measurement is gained by maintaining the type and number of malicious actions committed per service provider. The main aim of tier III is to monitor any wrong actions performed by a service provider. The three-tiers evaluation is performed using the proposed mathematical equations and algorithms. The output results of each tier of the three-tiers are inserted as an input to the buildup overall trust FIS, which provides a total trust value and status for a service provider per computational interval. This provides a full representation of a service provider abidance to the SLA contract, processing performance and violations committed during its service provisioning in the MEC paradigm. This paper is organized as follows; 'Literature Review and Related Work', introduces the literature review and related work. 'Proposed Integrated Three-Tier Trust Management Framework', presents the integrated three-tier trust management framework: Phase I. The three-tiers integration using fuzzy logic is detailed in 'Phase II: Three-Tier Integration using Fuzzy Logic'. 'Simulation Results and Discussion', shows the simulation results. Finally, the conclusion and future work are discussed in 'Conclusion and Future Work'.

## LITERATURE REVIEW AND RELATED WORK

Many researches had developed various trust evaluation schemes to assess service providers' provisioned services and behavior in different edge computing (EC) paradigms. This is in an attempt to improve service providers' quality of service (QoS) and decrease

**Table 1 Related work.**

| Ref. model | Domain | Parameters measured | Discussion |
|---|---|---|---|
| (*Monir, AbdelKader & EI-Horbaty, 2019*) | MEC | SLA was evaluated by computing users' opinion in service provider's processing cost, storage, maintenance and execution time. | Trust evaluation results were totally dependent upon service users' feedback opinion, which may led to less reliable trust results. |
| (*Ma & Li, 2018*) | EC | Trust was measured by evaluating deployed data security and privacy mechanisms in terms of resource identity, performance and quality of service. | Trust updating and sharing was not addressed, which weakens the trust evaluation efficiency of the model. |
| (*Deng et al., 2020*) | MEC | A reputation-based trust evaluation model and management for service providers was introduced that measured trust in terms of identity verification, deployed hardware capabilities (CPU, memory, disk, online time) and behavior. | Trust results were derived from service consumers' previous interactions' ratings. Unfortunately, such users' ratings may not be trustworthy enough. |
| (*Ruan, Durresi & Uslu, 2018*) | MEC | Service provider's trustworthiness is measured according to its performance per transaction with a service user. A degree of confidence measure is associated accordingly that shows user expectation of service provider future behavior. | The model depended on users' ratings, who could have different perspectives which may negatively affect trust evaluation accuracy. Monitoring and comparing such ratings in user-provider relationships is time consuming and may produce redundant data. |
| (*Khan, Chan & Chua, 2018*) | CC | Service providers' quality of service was evaluated in terms of service availability, response time and throughput. | Fuzzy rules were used to predict future behavior of a cloud service provider. The model helped service users in their service cost estimation. |
| (*Akhtar, 2014*) | CC | Service provider performance was evaluated in terms of infrastructure (response time and resource utilization with respect to the number of users) and application performance (in terms of; response time to a user, volume of data linked and processing migration). | Service provider performance evaluation was computed using fuzzy logic. Results managed to conclude the service provider performance level. |

losses emerging due to malicious actions performed over the network. This in return, will increase service users' trust and dependency EC resources (*Jhaveri et al., 2018*). Table 1 discusses some of the related work.

The abovementioned attempts measured trust considering different parameters, yet there isn't a unified service provider trust evaluation framework that integrates all major attributes together. Such main attributes are; SLA compliance degree, processing performance level and violations measurement of service providers' provisioned services in the MEC network.

# PROPOSED INTEGRATED THREE-TIER TRUST MANAGEMENT FRAMEWORK

The proposed framework aims to measure service provider trust value, considering various attributes. The model is built up of two phases, as shown in Fig. 1. Phase I constitutes three main tiers: Tier I evaluates service provider's SLA compliance degree, Tier II computes the processing performance value of a service provider, while Tier III measures the violations committed by a service provider during its processing in the MEC network. Phase II integrates the results of the three tiers in order to gain an overall trust value and status of a service provider using fuzzy logic concept.

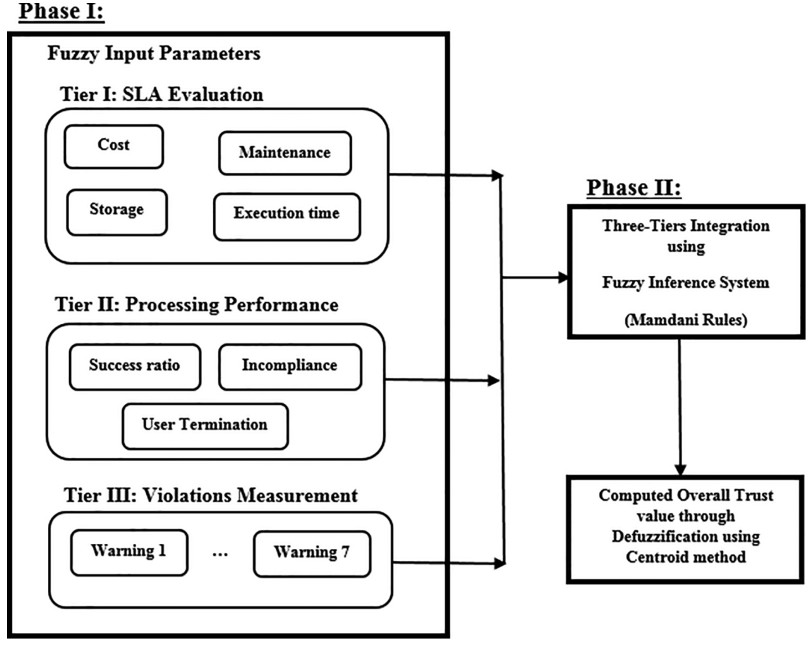

**Figure 1 FIS input variables membership function.**

## Acting protocol entities

- Service User "$SU_j$": $j^{th}$ service user, is a user requesting a certain job to meet its computational needs. $SU_j$ is represented by two attributes {service user unique id and name}.
- Service Provider "$Sr_i$": $i^{th}$ service provider, could be an ordinary provider or an organization supplying computational services to users. $Sr_i$ is represented by three attributes {service provider unique id, service provider name and offered service type}.
- Cloud Broker: acts as an intermediate entity to match between a service user seeking a suitable service provider. A cloud broker is considered as a semi-trusted entity.
- Cloud Service Manager (CSM): is regarded as a fully trusted authorized party in the MEC network. CSM is responsible to perform, regulate and audit trust computation process for service providers in the MEC environment (*Felix & Ricardo, 2012*). CSM can exchange computed trust values of service providers within its coverage range with other CSM, in case requested. CSM also provides secure storage of trust computed results of service providers.
- Network Provider: is responsible for registering a service user, provider and cloud broker to the MEC network. It also handles network communications between all of the above entities.

## List of assumptions

The proposed trust framework considers the below assumptions:

- Trust computation is handled by a fully trusted third party like CSM.

- In case service provider1 sends part or all of user's required task to service provider2 for processing, known as process migration, service provider1 is totally responsible for user's data security. Service provider1 should also inform the user of this attempt, and offer the relative guarantee to ensure service user's data security, privacy and integrity.
- A service provider could own one or more platform that offers one or more different service type.
- Each service type of a service provider is evaluated independently regardless it's the same service provider.
- There are three main jobs requested over the MEC network: 1-processing, 2-storage, 3-both of them, known as Job_type {Job1, Job2, Job3} respectively.

## Phase I: proposed three-tier trust evaluation framework

Phase I constitutes of three tiers trust evaluation: Tier I—Service level agreement evaluation, Tier II—Processing performance evaluation and Tier III—Violations measurement. In each tier, several parameters are evaluated to gain the tier trust value, as shown in Fig. 2.

Tier I trust results are gained by service user rating of SLA, upon process completion. Tier II is computed by evaluating the processing performance of a service provider, whereas Tier III provides a violations measurement and warnings received by a service provider. The three-tier computation is performed per "$n$" computational interval as described in the below subsections.

### Tier I—service level agreement evaluation

A service level agreement, is an agreement placed between a service provider and user, which states the job type requested by the service user and it's agreed upon computational conditions. Assume that the requested job is one of the previously mentioned three job types; known as "a", and the total number of requested jobs received by $i^{th}$ service provider, is referred to as "A". The main conditions mentioned in an SLA should be standard for all SLAs, maintained by both parties and eligible for judgment if needed (*Chong et al., 2013b*). Assume that all SLAs contain four major conditions; computational cost ($SC_{ia}$), required computational storage capacity GB/TB ($SS_{ia}$), computational maintenance duration h/min ($SM_{ia}$), and agreed computational execution time h/min be ($SE_{ia}$), (*Monir, AbdelKader & EI-Horbaty, 2019*). Upon job completion, a service user performs a compulsory rating process, to rate the service provider compliance to the four agreed upon conditions according to its own job execution experience. Assume that each of the above four major SLA components are rated as "r", shown in Table 2.

The abovementioned values could be adjusted according to own perspective.

Such ratings of computational cost, storage, maintenance and execution time ($SC_{ia}\_R$, $SS_{ia}\_R$, $SM_{ia}\_R$, $SE_{ia}\_R$) respectively, reflects user degree of satisfaction/dissatisfaction and compliance of a service provider against the agreed upon SLA conditions. Assume that the total number of rated SLAs, equals the total number of accepted processes "A" by $i^{th}$

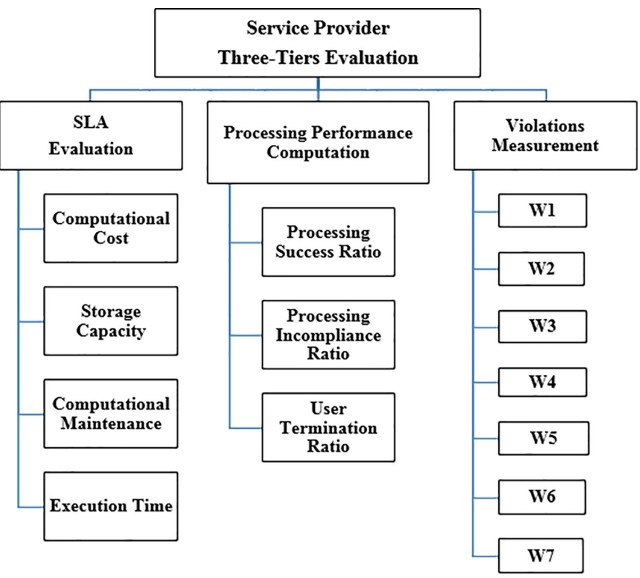

**Figure 2  Phase I: three-tiers trust computation attributes.**

**Table 2  SLA components rating & warnings.**

| Variable | Dissatisfaction rate | Satisfaction rate | Consequences of dissatisfaction rate > Threshold SLA drawback warnings |
|---|---|---|---|
| $SC_{ia}\_R$ | $0 \leq r \leq 2$ | $3 \leq r \leq 5$ | $W1\_Sr_i$ issued |
| $SS_{ia}\_R$ | $0 \leq r \leq 2$ | $3 \leq r \leq 5$ | $W2\_Sr_i$ issued |
| $SM_{ia}\_R$ | $0 \leq r \leq 2$ | $3 \leq r \leq 5$ | $W3\_Sr_i$ issued |
| $SE_{ia}\_R$ | $0 \leq r \leq 2$ | $3 \leq r \leq 5$ | $W4\_Sr_i$ issued |

service provider. Let the computed rated SLA value be $SLA_{ia}\_R\_value$, were it ranges between "0" and "20", given by Eq. (1), is as follows:

$$SLA_{ia}\_R\_value \, (SC_{iaR} + SS_{iaR} + SM_{iaR} + SE_{iaR}) * 5 \tag{1}$$

Eq. (1) is multiplied by five, to get the result in percentage form. Thus, the average computed SLAs value, "$SLA_{average}$", is calculated by;

$$SLA_{average} = \sum_{a=1}^{A} \frac{SLA_{ia}\_R\_value}{A} \tag{2}$$

SLAs average is computed per service provider as given in Eq. (2), by the CSM. This guarantees SLA evaluation results credibility. Noting that, each requested job is given a separate SLA, even if it's requested by the same service user, and performed by the same service provider. However, each job could have different computational requirements.

On the other hand, a certain threshold value is set for each dissatisfaction rated component. The dissatisfaction rate is computed per $i^{th}$ service provider for each component per "n" computational interval. In case the dissatisfaction rate of any

component had exceeded the predefined threshold for this component, a warning is issued for $i^{th}$ service provider as shown in Table 2. This alerts service providers for any dissatisfactory results gained for the SLA components in order to enhance their processing capabilities in this component.

### Tier II—processing performance computation

The processing performance of $i^{th}$ service provider "$P_i$", refers to the value of successful processes accomplished by a service provider in "n" computational interval. Assume process "a" ends, either as successful or incomplete. A complete process implies that the service provider had abided to all the four processing conditions and is referred to as process success, "$PS_i$". On the other hand, an incomplete process could be a result of one of the below three states:

1. a service provider had started the job processing but didn't complete it within its agreed conditions,
2. a service provider didn't start the job processing though accepted the job,
3. a service provider had started the job processing and is proceeding within its agreed upon conditions, and didn't exceed its process execution time ($SE_{ia}$). However, the service user wishes to terminate the job processing transaction.

States 1 and 2, are recommended as incomplete job processing, known as processing incompliance "$PI_i$" by a service provider. State 3 is referred to as user termination case, referred as "$UT_i$". Tier II evaluates the processing performance of $i^{th}$ service provider, in terms of:

- Average processing success ratio "$APS_i$": is considered as the number of successful processes $PS_i$ implemented by $i^{th}$ service provider, divided by the total number of accepted processes "A", as shown in Eq. (3):

$$APS_i = \frac{PS_i}{A} \tag{3}$$

- Average processing incompliance ratio "$API_i$": is the number of accepted processes by a service provider but failed to perform or complete them "$PI_i$", is computed by Eq. (4);

$$API_i = \frac{PI_i}{A} \rightarrow \text{ States 1 \& 2} \tag{4}$$

- Average user termination ratio "$AUTR_i$": where a service user feels dissatisfied for any reason, and decides to terminate its computational transaction, as represented in state no. 3. This is known as user termination ratio, "$UTR_i$", and computed by Eq. (5);

$$AUTR_i = \frac{UTR_i}{A} \rightarrow \text{ State 3} \tag{5}$$

**Table 3 Tier II imposed warnings.**

| Tier II-variables | Processing performance drawbacks warnings |
|---|---|
| $API_i$ > Threshold | W5_$Sr_i$ issued |
| $AUTR_i$ > Threshold | W6_$Sr_i$ issued |

The processing performance $P_i$ of $i^{\text{th}}$ service provider is gained by Eq. (6);

$$P_i = APS_i \tag{6}$$

where $0 \leq P_i \leq 1$ for "$n = 1$" computational interval

Eq. (6), shows the processing performance degree of a service provider in the MEC environment.

A predefined threshold is set for each of processing incompliance and user termination ratio values. In case a service provider computed results of these two parameters had exceeded this threshold within "$n$" computational interval, a relative warning is issued for $i^{th}$ service provider as shown in Table 3. This is to alert $i^{th}$ service provider for such incidents.

### Tier III: violations measurement

Tier III proposes two algorithms: 1—complain and evidence algorithm, 2—violations measurement algorithm. Data privacy leakage (*Talal & Quan, 2015*), incidents occurring by service providers are monitored through the "complain and evidence" algorithm as described in "Data privacy leakage complaint algorithm". On the other hand, the violations measurement algorithm presented in "Violations Measurement Computation", counts the warnings gained by a service provider per "$n$" computational interval, to gain its trust value in Tier III.

**(A) Data privacy leakage complaint algorithm**

A service provider is responsible for service user data privacy while a user is provisioning provider's services or applications, by installing the necessary data protection mechanisms. However, a service user may face a situation where it discovers that its own private data had been routed by a service provider, without prior permission for doing so, this hinders user's data privacy (*Deepa et al., 2020*; *Talal & Quan, 2013*). A data privacy leakage is known as a security threat, where it can lead to cyber-attacks, social issues, or cause user robbery by different means (*Javed et al., 2021*; *Iwendi et al., 2020*; *Vasani & Chudasama, 2018*). To monitor such service provider actions, a service user sends a complain message "$M_{DPR}$" to the CSM, against the suspected service provider including an evidence for this incident. The complaint message "$M_{DPR}$" should include:

1. screenshot of service user data appearing in unknown platform for the user;
2. service user data ownership evidence (could be a previous email sent from the user to the respective service provider including this data);
3. transaction SLA including $Sr_i$ & $SU_j$.

| | |
|---|---|
| | **Algorithm name:** Data Privacy Leakage. **Description:** Service user complaints of data privacy leakage against service provider. Executed by CSM. |
| 1. | **Input:** $M_{DPR}$ |
| 2. | **Output:** incident1_$Sr_i$ or W7_$Sr_i$ |
| 3. | CSM: |
| 4. | $SU_j \rightarrow CSM$: $M_{DPR}$ |
| 5. | *where* $M_{DPR}$ = (screenshot of user data, user data ownership evidence, SLA of transaction) |
| 6. | **CSM:** *verifies* $M_{DPR}$ |
| 7. | *if* ($M_{DPR}$ is approved) *then* |
| 8. | *issue* incident1_$Sr_i$ |
| 9. | *store* incident1_$Sr_i$ |
| 10. | *count* incident1_$Sr_i$ ++ |
| 11. | **CSM$\rightarrow Sr_i$:** incident1_$Sr_i$ |
| 12. | **CSM:** *checks* |
| 13. | *if* incident1_$Sr_i \geq 2$ then |
| 14. | *issue* W7_$Sr_i$ |
| 15. | *store* W7_$Sr_i$ |
| 16. | **CSM$\rightarrow Sr_i$:** W7_$Sr_i$ |
| 17. | *endif* |
| 18. | *else* |
| 19. | ($M_{DPR}$ is disapproved) |
| 20. | *exit* () |
| 21. | *endif* |
| 22. | *end* |

**Figure 3** Data privacy leakage complaint algorithm.

**Table 4 Warning number, reason and "θ" decreasing factor.**

| Warning name | Reason | "θ" decreasing factor | Tier |
|---|---|---|---|
| W1_$Sr_i$ | Exceeded cost dissatisfaction threshold. | $\theta W1 = 0.1$ | I-SLA |
| W2_$Sr_i$ | Exceeded storage dissatisfaction threshold. | $\theta W2 = 0.1$ | |
| W3_$Sr_i$ | Exceeded maintenance dissatisfaction threshold. | $\theta W3 = 0.1$ | |
| W4_$Sr_i$ | Exceeded agreed computational execution time dissatisfaction threshold. | $\theta W4 = 0.1$ | |
| W5_$Sr_i$ | Exceeded processing incompliance threshold. | $\theta W5 = 0.2$ | II-Processing performance |
| W6_$Sr_i$ | Exceeded user termination ratio threshold. | $\theta W6 = 0.1$ | |
| W7_$Sr_i$ | Data privacy leakage incident. | $\theta W7 = 0.3$ | III-Violations Measurement |

The mentioned conditions in the complaint message, "$M_{DPR}$" are investigated by the CSM for verification. If the CSM investigation results are approved to be true against $i^{th}$ service provider, this incident is considered as incident 1 and an alarm message "incident1_$Sr_i$" is sent to the accused service provider accordingly, as shown in Fig. 3. Incident 1 is counted and stored by the CSM.

In case a data privacy leakage complaint had been submitted by a different service user, but against the same service provider, CSM investigates the case. If CSM approves the complaint message to be true as mentioned previously, CSM considers these incidents as data privacy leakage attack, where the accused service provider is penalized by sending to it warning no. 7, "W7_$Sr_i$". On the other hand, such warning degrades the trust value for the accused service provider as shown in Table 4. This is to avoid recurrent occurrence of such attacks in the future by malicious service providers.

The data privacy leakage complaint algorithm presented in Fig. 3, traces and counts data leakage actions committed by $i^{th}$ service provider in "n" computational interval, with

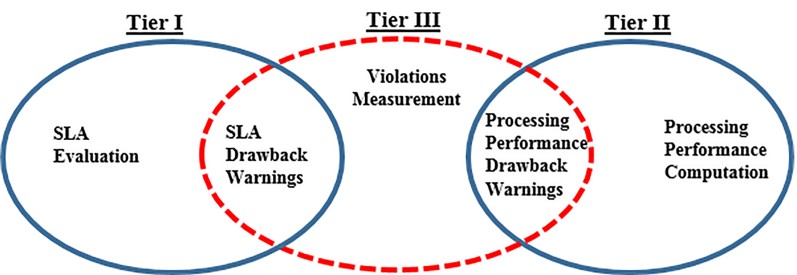

**Figure 4** Relationship between the proposed Three-Tier framework warnings.

different users. These incidents and warning are being stored in a historical database by the CSM per $i^{th}$ service provider.

**(B) Violations measurement computation**

The violations measurement computation algorithm measures the warnings imposed on $i^{th}$ service provider in tiers I and II, (during SLA evaluation and processing performance computation), and in case of a data privacy leakage incident. The relationship between the three-tier framework warnings is shown in Fig. 4.

Each warning type is given a number, from 1 to 7, as shown in Table 4. A decreasing factor "θ" is given for warnings imposed on $i^{th}$ service provider, which is computed according to the number and reason of warning. Noting that "θ" is a changing variable, that could be assigned different values according to own perspective. Assume the total warnings value for $i^{th}$ service provider computed in Tier III, be "$V_i$", which is calculated by Eq. (7):

$$V_i = \sum \theta W1, \ \theta W2, \ \theta W3, \ \ldots, \ \theta W7 \qquad (7)$$

where $0 \le V_i \le 1$

Given that "$V_i$" gives the total warnings value, hence, the violations measurement "$TV_i$", is computed by;

$$TV_i = \ 1 \ - \ V_i \qquad (8)$$

The violations measurement of $i^{th}$ service provider is computed in Tier III, as shown in Fig. 5, by the CSM. In this algorithm, CSM checks whether $i^{th}$ service provider had received any warnings (W1_$Sr_i$, …, W7_$Sr_i$). If $i^{th}$ service provider received any warning, it computes its relative "θ" and "$V_i$" value by Eq. (7) accordingly. "$TV_i$" is computed by Eq. (8), to gain the violations measurement for $i^{th}$ service provider.

The violations measurement algorithm presented in Fig. 5, checks the warnings received by $i^{th}$ service provider, to compute its total warnings value "$V_i$". Hence, the total warnings value is deducted from "1", to gain its violations measurement "$TV_i$". In case a service provider didn't receive any warnings, this service provider gains the full value of the violations measurement "$TV_i$", which is equal to "1".

| | |
|---|---|
| **Algorithm name:** Violations measurement. **Description:** Warnings computation. Executed by CSM. | |
| 1. | **Input:** $W1\_Sr_i, \ldots, W7\_Sr_i$ |
| 2. | **Output:** $TV_i$ |
| 3. | CSM: |
| 4. | *If* $W1\_Sr_i \mid W2\_Sr_i \mid W3\_Sr_i \mid W4\_Sr_i \mid W5\_Sr_i \mid W6\_Sr_i \mid W7\_Sr_i$ is true *then* |
| 5. | *compute* $V_i =$ // equation (7) |
| 6. | *compute* $TV_i =$ // equation (8) |
| 7. | *else* |
| 8. | $TV_i = 1$ |
| 9. | *exit* () |
| 10. | *endif* |
| 11. | *return* $TV_i$ |
| 12. | *end* |

**Figure 5  Violations measurement algorithm.**     

## Summary

The proposed three-tier protocol in Phase I, aims to maintain the trust value of $i^{th}$ service provider considering three main attributes; SLA compliance degree, processing performance level and violations measurement. This is in an attempt to optimize trust results credibility. In Phase II, an integration of the three-tier results is provided to gain an overall trust value of $i^{th}$ service provider, building up the whole framework. This disseminates service providers' processing performance and pervious interactions in the MEC network.

## PHASE II: THREE-TIER INTEGRATION USING FUZZY LOGIC

The overall trust value of $i^{th}$ service provider is an aggregate value of the computed end results of each of the above three tiers (SLA evaluation, processing performance computation and violations measurement). Hence, $i^{th}$ service provider overall trust value is computed using the developed fuzzy inference system (FIS) named "Overall_Trust", as shown in Fig. 1 and described in 'Using Fuzzy Logic for Trust Computation in Mobile Edge Computing' and 'Integration of Tiers I, II and III Results using Fuzzy Logic'.

### Using fuzzy logic for trust computation in mobile edge computing

Fuzzy logic is a form of artificial intelligence, were the input parameters are given to a fuzzy inference system as unclear or uncertain information, denoting partial truth of a parameter (*Nagarajan, Selvamuthukumaran & Thirunavukarasu, 2017*). This is in contrast of Boolean logic, which belongs to discrete numbers, either 0 or 1. Such input values range between 0 and 1 and are placed in membership functions to distinguish each range of values, known as fuzzification process (*Sule et al., 2017*). If-then-else rules are set to the fuzzy inference rule base editor, in MATLAB, in order to allocate each range of input values to a specific output decision. The output result is converted into a crisp value known as defuzzification process. Fuzzy logic concept has great advantages, like flexibility, fast response time, low cost and logical reasoning. For these reasons, fuzzy logic was chosen to compute service providers overall trust value and status in this framework, since MEC is a

**Peer**J Computer Science

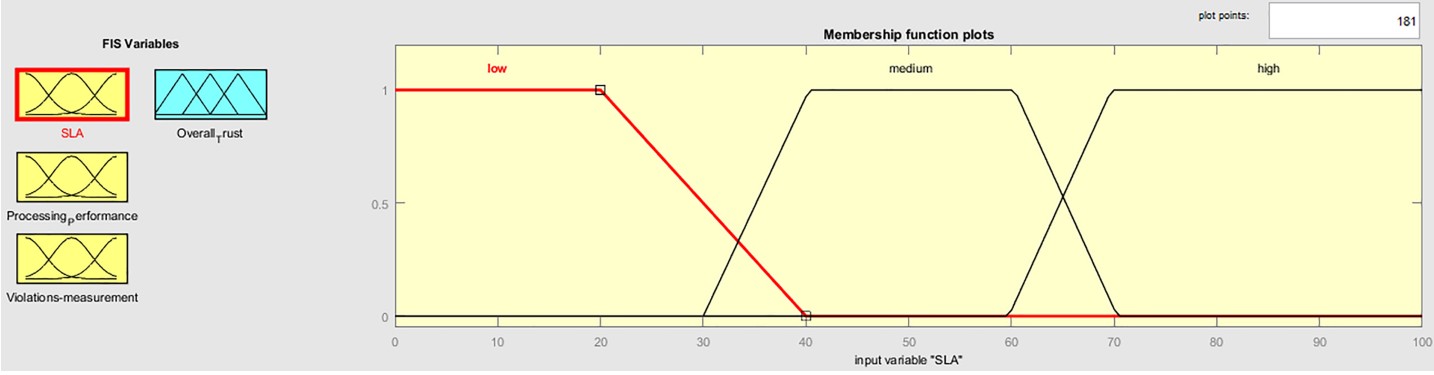

**Figure 6 FIS input variables membership functions.**

| Table 5  System design and structure. | | | | | | | |
|---|---|---|---|---|---|---|---|
| **FIS Name** | Overall Trust | | | | | | |
| **FIS Type** | Mamdani | | | | | | |
| **Rules** | 21 | | | | | | |
| **Defuzzification type** | Centroid | | | | | | |
| | **Input** | | | **Output** | | | |
| **No. of** | 3 | | | 1 | | | |
| **Name** | SLA | | | Overall Trust | | | |
| | Processing performance | | | | | | |
| | Violations measurement | | | | | | |
| **No. of membership functions** | 3 | | | 4 | | | |
| **Membership function Type** | Trapezoidal | | | Triangular | | | |
| **Membership function name** | Low | Medium | High | Low | Medium | High | Excellent |
| **Range** | 0-40 | 30-70 | 60-100 | 0-30 | 30-60 | 60-90 | 90-100 |

highly dynamic environment. Service providers trust computation helps service users during their service provider selection, and balances between offered services and cost.

## Integration of tiers I, II and III results using fuzzy logic

Fuzzy logic concept is used to integrate the end results of Tiers I, II, and III of Phase I, in order to gain an overall trust value for $i^{th}$ service provider, during Phase II computation, as shown in Fig. 1. The fuzzy logic toolbox in MATLAB was used to develop the "overall trust" system. Trapezoid-cure shape was used to represent each of the three fuzzy inputs of the three tiers, (SLA evaluation, processing performance computation, violations measurement), and their relative fuzzy membership functions, as shown in Fig. 6. Membership functions input values range between [0,100], which are converted to fuzzy linguistic input variables, forming three fuzzy sets (low, medium, high), during the fuzzification process, as presented in Table 5.

Centroid defuzzification was performed to obtain a crisp overall trust value for $i^{th}$ service provider. The triangular-shape curve represents the output value membership functions (low, medium, high, excellent), as shown in Fig. 7. $i^{th}$ service provider overall

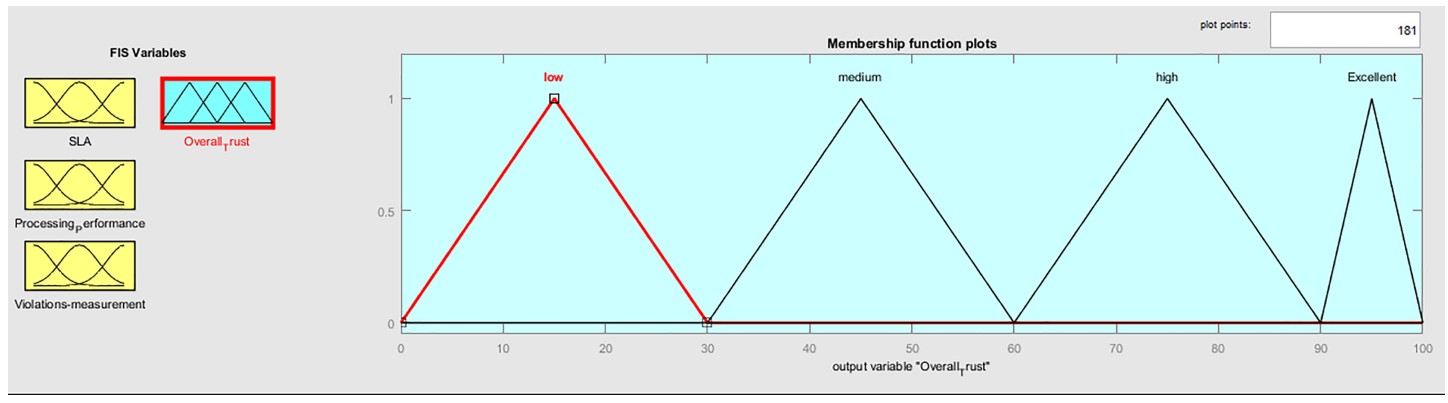

**Figure 7 FIS output membership functions.**               

**Table 6 Fuzzy inference system rules.**

| Rule No. | Fuzzy input variables $SLA_{average}$ | Defuzzified output $P_i$ | $WT_i$ | Overall Trust |
|---|---|---|---|---|
| 1. | L | L | L | Low |
| 2. | L | L | M | Low |
| 3. | L | M | L | Low |
| 4. | M | L | L | Low |
| 5. | L | L | H | Low |
| 6. | L | H | L | Low |
| 7. | H | L | L | Low |
| 8. | M | M | M | Medium |
| 9. | M | M | L | Medium |
| 10. | M | L | M | Medium |
| 11. | L | M | M | Medium |
| 12. | M | M | H | Medium |
| 13. | M | H | M | Medium |
| 14. | H | M | M | Medium |
| 15. | H | H | M | High |
| 16. | H | M | H | High |
| 17. | M | H | H | High |
| 18. | H | H | L | Medium |
| 19. | H | L | H | Medium |
| 20. | L | H | H | Medium |
| 21. | H | H | H | Excellent |

trust value belongs to one of the four fuzzy output sets. Twenty-one inference rules were added to the Mamdani inference system, to compute the overall trust status of $i^{th}$ service provider, considering its SLA evaluation, processing performance computation and violations measurement. Each rule, used AND logical operator, to co-relate between input and output variables, as given in Table 6.

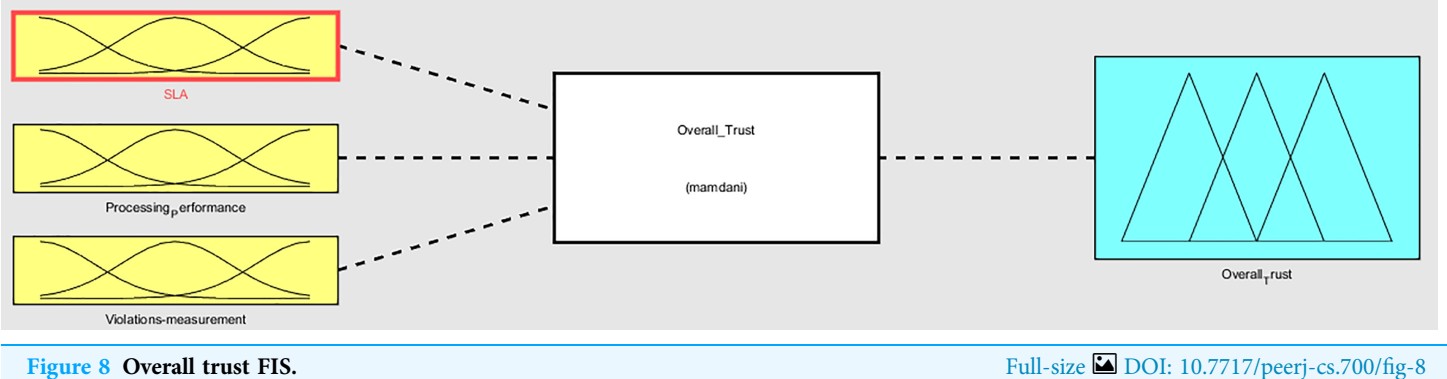

**Figure 8 Overall trust FIS.**

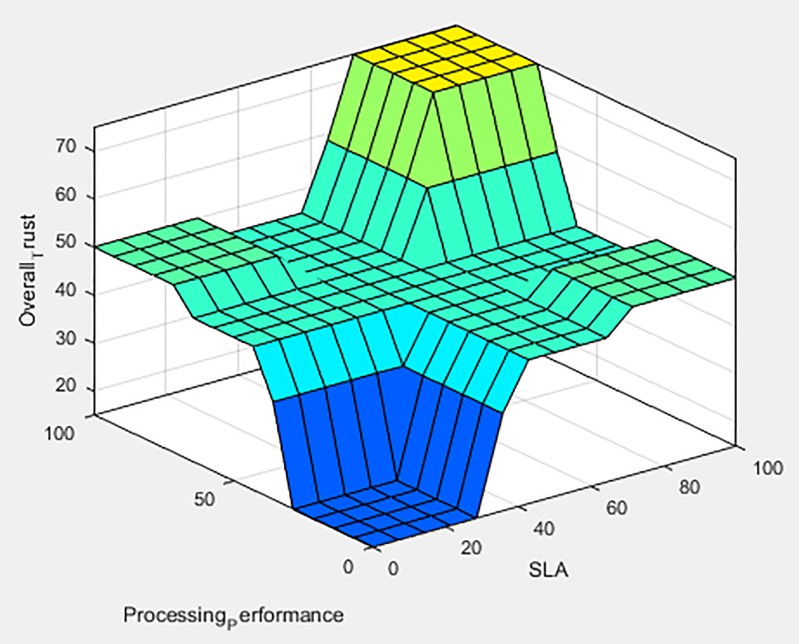

**Figure 9 Surface view of processing performance and SLA evaluations verses overall trust.**

The fuzzification of the three-tiers values participate in forming the overall trust fuzzy inference system, which computes $i^{th}$ service provider overall trust in the MEC network, as presented in Fig. 8. A smooth surface is shown in Fig. 9, while comparing the processing performance and SLA evaluations against the overall trust value.

## Phase II: summary

The main aim of Phase II, is to provide an overall trust value for a service provider in the MEC network. This is done by gaining the value of service provider's successful interactions and the violations degree measurement, while maintaining service users'

opinion of the SLA. Through the fuzzy logic concept, the three tiers' end results were integrated to produce $i^{th}$ service provider overall trust value and status.

## SIMULATION RESULTS AND DISCUSSION

The integrated three-tier trust management framework simulation was performed using Matlab R2017a. Simulation results are shown in 'Integrated Three-Tier Trust Management Framework Simulation Results'. A comparison with previous protocols is presented in 'Comparison with Previous Protocols'. 'The Intergrated Three-Tier Trust Management Framework Achievements' depicts the proposed framework achievements.

### Integrated three-tier trust management framework Simulation Results

A MATLAB-based simulation was developed to show Phase I and II results of the integrated three-tier trust management framework. Random number generation was used for the input values of each of: SLA rating components; number of received processes (successful, failed and user termination ratio); and privacy leakage incident. The overall trust fuzzy inference system was developed using MATLAB fuzzy logic designer tool. However, each of the simulated service providers was forced a specific range of values, to validate the developed system equations and twenty-one fuzzy inference system rules set, in various conditions. The overall trust FIS contains three input membership functions. Each membership function presents the computed results of each tier (SLA, processing performance and violations measurement).

**System setup:**

1. Assume five different service providers' cases studies.
2. Each service provider initial trust value is "0".
3. Three-tier computation was performed in n = 1 computational interval.
4. All service providers received the same number of job requests, in one job_type {Job3}.
5. Hardware PC configuration was; core i7, RAM 6 GB and hard disk 1 Tera.

The three-tier simulation results of Phase I for the five service providers are shown in Table 7. These results are shown for $i^{th}$ service provider, upon the completion of "$n$" computational interval. Phase I presents detailed computational results and points of enhancement for each service provider per tier. The warning methodology helps a service provider to know its points of weakness.

Phase II simulation results are shown in Table 8. The overall trust FIS computes the trust value and status per service provider in one of the predefined job types. In case a service user is seeking the processing of job3, overall trust value and status of the available service providers in this job type is shown in Table 8. However, a service user could reveal the three-tier results, with the overall trust value and status of the five service providers for serious selection, as shown in Fig. 10.

Fig. 10 gives a full representation for each of the simulated five service providers processing performance in the MEC network.

**Table 7 Phase I simulation results: three-tier results and enhancement recommendations.**

| Service Provider | SLA | PP | Violations | Warnings issued | Enhancement Recommendation | | |
|---|---|---|---|---|---|---|---|
| | | | | | SLA | PP | Violations |
| Sr1 | 80.6 | 87.8 | 83.3 | W1_$Sr_i$ | √ -SC | | |
| Sr2 | 50 | 50 | 50 | W2_$Sr_i$, W4_$Sr_i$, W6_$Sr_i$, W7_$Sr_i$ | √ -SS -SE | √ -UTR | √ -DPR |
| Sr3 | 20.3 | 22.1 | 23 | W2_$Sr_i$, W3_$Sr_i$, W4_$Sr_i$, W5_$Sr_i$, W6_$Sr_i$, W7_$Sr_i$ | √ -SS -SM -SE | √ -PI -UTR | √ -DPR |
| Sr4 | 48.2 | 56.3 | 86 | W2_$Sr_i$, W6_$Sr_i$, | √ - SS | √ -UTR | |
| Sr5 | 80.6 | 50.9 | 23.9 | W5_$Sr_i$, W6_$Sr_i$, W7_$Sr_i$ | | √ -PI -UTR | √ -DPR |

**Table 8 Phase II simulation results: service provider overall trust value and status.**

| Service provider | Overall trust value | Overall trust status |
|---|---|---|
| Sr1 | 95 | Excellent |
| Sr2 | 45 | Medium |
| Sr3 | 15 | Low |
| Sr4 | 45 | Medium |
| Sr5 | 50 | Medium |

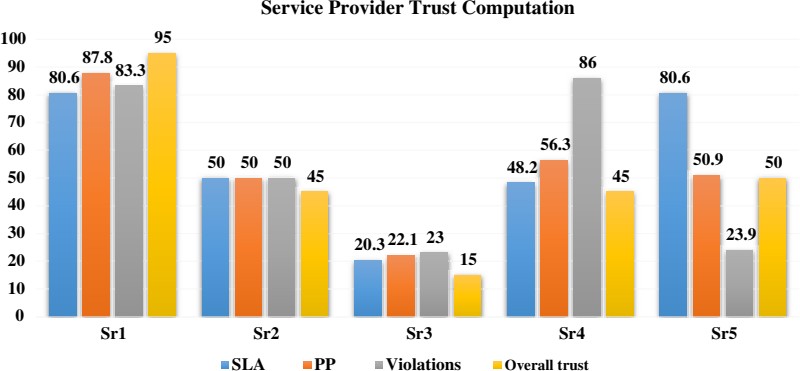

**Figure 10 Service providers overall trust results.**

## Comparison with previous protocols

A comparison between the proposed three-tier trust management framework evaluated parameters, and previous protocols measured parameters in cloud computing and mobile edge computing is presented in Table 9.

**Table 9 Comparison with previous protocols.**

| Ref.no. | Platform | Service provider evaluated parameters | | | | History Capturing | Trust Evaluation Methodology | Discussion and limitations |
|---|---|---|---|---|---|---|---|---|
| | | SLA | Processing Performance | Violations measurement | Other | | | |
| (*Akhtar, 2014*) | CC | | √ | | Infrastructure performance | | Fuzzy logic-based rules | Measured limited parameters for each performance variable. |
| (*Khan, Chan & Chua, 2018*) | CC | √ | | | | | System training using Fuzzy logic-based rules | Didn't evaluate other parameters, like processing success rate, data security or privacy measures taken by a service provider. |
| (*Monir, AbdelKader & EI-Horbaty, 2019*) | MEC | √ | | | | √ | -User feedback opinion | Only evaluated SLA by considering service users' feedback opinion, which may affect trust results credibility. |
| (*Deng et al., 2020*) | MEC | | √ | | Identity management and hardware capabilities | | -User feedback opinion | The hardware capabilities are claimed by the service provider, which doesn't ensure their credibility. Processing performance evaluation depends only on users' opinion, which may be biased. The model doesn't encounter threats such as bad mouthing or collusion attacks. |
| Integrated Three-Tier Trust Management Framework | MEC | √ | √ | √ | | √ | -User feedback opinion -Computation-based with the aid of fuzzy logic concept | Service user trust value should be measured. Transaction cost should be considered to eliminate false transactions rating. |

As shown in Table 9, the proposed framework had successfully measured various service performance parameters for service providers, with a warning and history capturing methodology, in comparison to previous works.

**The intergrated three-tier trust management framework achievements**

The proposed integrated three-tier trust management framework achieves the following:

- Service provider assessment based on its behavior during its interactions on the MEC network according to its predefined job type and not by the quantity of hardware or software resources it possess, or job type offered.
- Service providers' awareness of their faulty points, while displaying the degree of improvement needed per component.

- Future prediction for service provider performance competency per tier.
- Malicious service provider detection and their points of attack through the violations measurement, warnings mechanism and the low trust fuzzy membership function.
- Trust evaluation with minimal human interaction to maximize trust results credibility.
- Dynamic trust computation for service providers, per "$n$" computational interval, considering history capturing mechanism.
- Helps in cost estimation, for example, if a service provider's service cost is high, this could be justified if its trust value is high in the three evaluated components. Service users' awareness and guidance during their service selection for credible and trustworthy service providers. Consequently, service users will gain confidence and increase their dependency on the MEC network services.
- Low computational trust evaluation time complexity. This is mainly due to: 1—the simplicity of the model equations, 2—history capturing makes trust results update easy, 3—the developed fuzzy inference system. Thus, the model can be used for large number of service providers in the MEC network, which is time efficient.
- Computational storage saving, were there is no data redundancy during trust computation.

## CONCLUSION AND FUTURE WORK

Finding credible service providers in a vast ambiguous environment like mobile edge computing had been a very hard task faced by service users. The proposed integrated three-tier trust management framework, measures the trust value of a service provider considering three main attributes; SLA compliance, processing performance and violations measurement, in Tiers I, II and III of Phase I. Tiers I, II and III constitutes of different evaluated parameters, which eliminates the opportunity of false ratings and collusion attacks by service users. However, the three-tier protocol shows service provider's points of weakness or strength per tier. The three-tier results are aggregated to give an overall trust value and status for a service provider, using the developed overall fuzzy inference system in Phase II. Trust evaluation in performed in a history capturing manner to ease trust updating process.

From a service user perspective, the three-tier protocol could also be beneficial, since service users may have different preferences, in spite they are requesting the same job type. For example, a service user may search for a service provider which will abide to its SLA, in terms of time compliance, for time critical operations, rather than its processing performance quality. This obviously helps service users and organizations in their selection for credible service providers to fulfil their computational demands.

The integrated three-tier framework computational simulation results show the trust value of a service provider in terms of SLA compliance, processing performance, and violations occurrence. The proposed warnings protocol, shows each service provider points of weakness, which supports its improvement. Finally, the results of the three-tier

framework integration reflects the overall trust value and status of a service provider in the MEC network.

In the future, we plan to measure service user trust value, which could be a factor multiplied by its rated value. Therefore, the higher this factor, the more likely this rating to be true, which increases the service provider computed trust results reliability. On the other hand, service user trust value can be considered as a filtering mechanism for falsified ratings. However, the transaction cost should be considered as an affecting weight, during trust computation. Usually transactions of high cost are rarely mentioned to be fake. The active work time of a service provider should be measured in comparison to its total subscription life time in the MEC network. This actually reflects dedicated operational service providers.

### Funding
This is work is fully funded by Merrihan B.M. Mansour. The funders had no role in study design, data collection and analysis, decision to publish, or preparation of the manuscript.

### Grant Disclosures
The following grant information was disclosed by the authors:
This is work is fully funded by Merrihan B.M. Mansour.

### Competing Interests
All authors declare that they have no competing interests.

### Author Contributions
- Merrihan B.M. Mansour conceived and designed the experiments, performed the experiments, analyzed the data, performed the computation work, prepared figures and/or tables, authored or reviewed drafts of the paper, and approved the final draft.
- Tamer Abdelkader performed the experiments, performed the computation work, prepared figures and/or tables, and approved the final draft.
- Mohamed Hashem analyzed the data, prepared figures and/or tables, and approved the final draft.
- El-Sayed M. El-Horbaty conceived and designed the experiments, authored or reviewed drafts of the paper, and approved the final draft.

### Data Availability
The code is available in the Supplemental Files.
The fuzzy inference rules, Phase I simulation results for the three-tiers (service level agreement, processing performance and violations measurement, including warnings and enhancement recommendations) and Phase II simulation results are available in the Tables.

## Supplemental Information

Supplemental information for this article can be found online at http://dx.doi.org/10.7717/peerj-cs.700#supplemental-information.

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
