# Peer review of "An integrated three-tier trust management framework in mobile edge computing using fuzzy logic"

_PeerJ Computer Science, doi:10.7717/peerj-cs.700_

## Round 0.1 · original submission · Major Revisions

Dear Dr. B.M.Mansour,

Thank you for your submission to PeerJ Computer Science.
It is my opinion as the Academic Editor for your article - An integrated three-tier trust management framework in mobile edge computing using fuzzy logic - that it requires a number of Major Revisions.

My suggested changes and reviewer comments are shown below and on your article 'Overview' screen.

Reviewer 1 ·

Basic reporting

This paper presents an integrated three-tier trust management framework that evaluates cloud service providers in three main domains; tier I-evaluates service provider compliance to the agreed service level agreement, tier II computes the processing performance of a service provider based on its number of successful processes, and tier III-measures the violations committed by a service provider during its processing in the MEC network. The three-tier evaluation is performed in phase I computation. In phase II, a service provider trust value and status are gained through the integration of the three tiers using the developed overall trust fuzzy inference system (FIS).

Experimental design

Satisfacatory

Validity of the findings

Satisfactory

Additional comments

1. Even though the motivations of the current work are clearly discussed, the contributions of the paper are not discussed. What are the main contributions of the current work?
2. The literature review section can be summarized as a table.
3. Some of the recent works on fuzzy systems, cloud computing and security such as the following can be discussed "Load balancing of energy cloud using wind driven and firefly algorithms in internet of everything, Big data for cybersecurity: Vulnerability disclosure trends and dependencies, Senti‐eSystem: A sentiment‐based eSystem‐using hybridized fuzzy and deep neural network for measuring customer satisfaction".
4. The main contribution of any paper is the proposed work. This section is very small in this paper. It has to be elaborated with discussion on he novelty and more detailed discussion on the proposed work.
5. The limitations and the future scope of the current work can be discussed in conclusion.
6. Also, the application of big data analytics in dealing with big data in the cloud can be added as a future direction as discussed in "A survey on blockchain for big data: Approaches, opportunities, and future directions".

·

Basic reporting

Many grammatical errors and typos are found (all the way from Abstract to Conclusion).

Experimental design

Comparison of the proposed work with existing work can be useful.

Validity of the findings

no comment

Additional comments

1. The flow of the introduction should be.. introduce the problem at high level, discuss about some of the existing solutions, identify the gap or scope of improvement, and then discuss in order to address the identified gaps what is the methodology we are using after that we have to list out the contributions
2. Literature review needs to be revised majorly and extended by including some latest articles. Comparison of different works can be summarized in a table.
3. Section 3 contains very less contents. Can it be merged with Section 4 along with Section 5?
4. Figures quality are poor.
5. Comparison of the proposed work with existing work can be useful.
6. Conclusion should be revised to write the concluded facts and should not reflect abstract.
7. The authors can discuss about the following relevant papers in their work:
Consumer Electronic Devices: Evolution and Edge Security Solutions
A Sequence Number Prediction Based Bait Detection Scheme to Mitigate Sequence Number Attacks in MANETs
A Trusted Social Network using Hypothetical Mathematical Model and Decision-based Scheme

Reviewer 3 ·

Basic reporting

In this paper, the authors presented an integrated three-tier trust management framework
that evaluates cloud service providers in three main domains. Below are some comments to improve the paper.

- Please highlight the contribution clearly in the introduction
- this paper lacks in Novelty of the proposed approach. The author should highlight the contribution clearly in the introduction and provide a comparison note with existing studies.
- Some Paragraphs in the paper can be merged and some long paragraphs can be split into two.
- The quality of the figures can be improved more. Figures should be eye-catching. It will enhance the interest of the reader.
- Figure 11, only the graph area should be added to the paper. remove grey borders. and the same for others.
- The background of figure "Figure 12 Service Providers Overall Trust results" should be white with font color black.


- Authors should add the most recent reference:
1) https://onlinelibrary.wiley.com/doi/abs/10.1002/spe.2846
2) https://ieeexplore.ieee.org/abstract/document/9068217
3) https://link.springer.com/article/10.1007/s11063-020-10414-5
- please give a proofread check to the paper.

Experimental design

- What are the computational resources reported in the state of the art for the same purpose?

Validity of the findings

- What are the evaluations used for the verification of results?
- Clearly highlight the terms used in the algorithm and explain them in the text.

---

## Round 0.2 · Minor Revisions

All the comments have been responded well and the reference list has been updated properly. The paper contains publishable contents, however, I notice that Figs, 6-10 are screenshots. In my opinion, it is not necessary to show the Matlab window (Figs. 6,7,9,10) while the figures in the window can be shown directly. Fig.8 can be explained in text, the image now is not given in high quality. A minor revision is recommended to fix them out.

---

## Round 0.3 · accepted · Accept

All the comments have been addressed well and the additional request in terms of image quality has been responded well. Based on the contribution and quality of the revised manuscript, i recommend accepting this paper.